# Tracking SARS-CoV-2 Spike Protein Mutations in the United States (January 2020—March 2021) Using a Statistical Learning Strategy

**DOI:** 10.3390/v14010009

**Published:** 2021-12-21

**Authors:** Lue Ping Zhao, Terry P. Lybrand, Peter B. Gilbert, Thomas R. Hawn, Joshua T. Schiffer, Leonidas Stamatatos, Thomas H. Payne, Lindsay N. Carpp, Daniel E. Geraghty, Keith R. Jerome

**Affiliations:** 1Fred Hutchinson Cancer Research Center, Public Health Sciences Division, Seattle, WA 98109, USA; 2Quintepa Computing LLC, Nashville, TN 37205, USA; terry.p.lybrand@vanderbilt.edu; 3Department of Chemistry, Vanderbilt University, Nashville, TN 37235, USA; 4Fred Hutchinson Cancer Research Center, Vaccine and Infectious Disease Division, Seattle, WA 98109, USA; pgilbert@fredhutch.org (P.B.G.); jschiffe@fredhutch.org (J.T.S.); lstamata@fredhutch.org (L.S.); lcarpp@fredhutch.org (L.N.C.); kjerome@fredhutch.org (K.R.J.); 5Department of Medicine, University of Washington School of Medicine, Seattle, WA 98195, USA; thawn@uw.edu (T.R.H.); tpayne@uw.edu (T.H.P.); 6Department of Global Health, University of Washington, Seattle, WA 98105, USA; 7Fred Hutchinson Cancer Research Center, Clinical Research Division, Seattle, WA 98109, USA; geraghty@fredhutch.org

**Keywords:** homology modelling, SARS-CoV-2, Spike protein, statistical learning, unsupervised learning, variants of concern, variants of interest, viral residue variant

## Abstract

The emergence and establishment of severe acute respiratory syndrome coronavirus 2 (SARS-CoV-2) variants of interest (VOIs) and variants of concern (VOCs) highlight the importance of genomic surveillance. We propose a statistical learning strategy (SLS) for identifying and spatiotemporally tracking potentially relevant Spike protein mutations. We analyzed 167,893 Spike protein sequences from coronavirus disease 2019 (COVID-19) cases in the United States (excluding 21,391 sequences from VOI/VOC strains) deposited at GISAID from 19 January 2020 to 15 March 2021. Alignment against the reference Spike protein sequence led to the identification of viral residue variants (VRVs), i.e., residues harboring a substitution compared to the reference strain. Next, generalized additive models were applied to model VRV temporal dynamics and to identify VRVs with significant and substantial dynamics (false discovery rate q-value < 0.01; maximum VRV proportion >10% on at least one day). Unsupervised learning was then applied to hierarchically organize VRVs by spatiotemporal patterns and identify VRV-haplotypes. Finally, homology modeling was performed to gain insight into the potential impact of VRVs on Spike protein structure. We identified 90 VRVs, 71 of which had not previously been observed in a VOI/VOC, and 35 of which have emerged recently and are durably present. Our analysis identified 17 VRVs ~91 days earlier than their first corresponding VOI/VOC publication. Unsupervised learning revealed eight VRV-haplotypes of four VRVs or more, suggesting two emerging strains (B1.1.222 and B.1.234). Structural modeling supported a potential functional impact of the D1118H and L452R mutations. The SLS approach equally monitors all Spike residues over time, independently of existing phylogenic classifications, and is complementary to existing genomic surveillance methods.

## 1. Introduction

Severe acute respiratory syndrome coronavirus 2 (SARS-CoV-2), the pathogen responsible for the global coronavirus disease 2019 (COVID-19) pandemic, is an RNA virus and thus prone to replication errors [1]. Replication errors that yield nonsynonymous amino acid (AA) substitutions, or nucleotide insertions or deletions that cause a frame shift and alter the subsequent coding sequence, can lead to a variety of outcomes. If the resulting mutations have detrimental effects on fitness, or if they have neutral effects on fitness and undergo stochastic extinction, variants harboring these mutations fail to become established in the population. However, mutations that confer a fitness advantage can rapidly become dominant in a population. For SARS-CoV-2, there are four classes of variant: variant being monitored (VBM), variant of interest (VOI), variant of concern (VOC), and variant of high consequence (VOHC). The CDC is closely monitoring ten VBMs (Alpha, Beta, Gamma, Epsilon, Eta, Iota, Kappa, 1.617.3, Mu, and Zeta) and two VOCs (Delta and Omicron) in the United States [2]. VOCs show specific attributes such as increased transmissibility [3,4,5,6], increased resistance to neutralization by antibodies elicited through natural infection [3,7,8,9], and/or increased resistance to neutralization by vaccine-elicited antibodies [8,10,11], and have already influenced vaccine development, evidenced by the current planning of clinical trials to test variant-adapted vaccines [12]. While no VOHCs have yet been identified, it remains possible that such variants—i.e., variants that can effectively evade natural or vaccine-induced immunity—may yet emerge [13,14]. The identification of VOHCs could necessitate the introduction of more stringent public health guidelines and/or spur further treatment and vaccine development.

Genomic surveillance is critical for tracking the emergence and spread of new variants. Such surveillance can be accomplished via a variety of approaches, such as phylogenic analysis [3,15]. In this approach, new viral sequences are classified to existing lineages identified by PANGO [16], subsets of samples with the same branches are identified, and variant frequencies are counted to identify new variants. The NextStrain methodology [17] can model dynamic changes in variant proportions, while an alternative approach aligns sequence data to a matrix of binary indicators for the presence of variants, and systematically evaluates each mutant as a potential variant [18]. Leveraging the analytic approach of single nucleotide polymorphisms (SNPs), variants have been identified by assessing linkage-disequilibrium [19] or similar SNP-based identification and analysis [20]. However, with the exception of the NextStrain methodology [3], these methods do not directly take into account sequence collection time, nor explicitly incorporate highly granular geographic information. Moreover, these methods take a holistic view of the viral genome. Thus, there is a need for complementary approaches for detecting and characterizing Spike mutations of potential public health importance that may be missed, or detected later, by existing genomic surveillance methods.

To meet this need, we describe a statistical learning strategy (SLS) using generalized additive models, unsupervised learning techniques, and single nucleotide polymorphism (SNP) methodologies for identifying and spatiotemporally characterizing viral residue variants (VRVs), a term we use to describe AA positions in the Spike protein where a mutation is significantly present in a given geographic area. The SLS method generates pertinent statistics for reproducible scientific inference and facilitates visual representation of results for intuitive interpretation. Using publicly available SARS-CoV-2 sequences from US COVID-19 cases that were not assigned to a VOI or VOC lineage, we apply our method to identify and spatiotemporally characterize VRVs in the Spike protein, within individual US states/territories. We also apply standard homology modeling methods to highlight individual AA mutations with the potential to impact Spike protein structure and/or function.

## 2. Results

### 2.1. Ab Initio Discovery of VRVs

We first applied the SLS method to identify VRVs separately in each state/territory (Appendix A). The decision to compartmentalize VRV discovery by state/territory was partially based on the fact that domestic travel restrictions have varied over the course of the pandemic, with nearly half of all states having imposed some type of interstate travel restriction [21], leading to the hypothesis that VRVs may follow state/territory-specific temporal dynamics. The identified VRVs showed a range of dynamic patterns across the different states/territories (Appendix A), exemplified by the five different trajectories taken by the V382, L452, T478, P681, and T732 VRVs in California (Figure 1A). The relative abundance of V382 started rising on day 250, exceeded 10% on day 259, and fell below 10% on day 275. L452 emerged on day 310, exceeded 10% on day 390, and exhibited a positive trajectory thereafter. Three other VRVs (T478, P681, T732) had similar trajectories to L452. Note that we denote the VRV of interest using the reference amino acid position alone (e.g., P681), given that some reference amino acids show point mutations to more than one amino acid (see examples below).

We refer to the combination of a VRV and a state/territory in which it was identified as a “geo-VRV”. A total of 267 geo-VRVs, consisting of combinations of 90 VRVs identified among the 52 state/territory classifications, were identified (Appendix A). Fifty-eight VRVs were only observed in one state/territory, whereas 32 were observed in two or more (Appendix A).

Unsupervised learning was then applied to organize the 267 geo-VRVs into 10 clusters (TP1 through TP10) (Figure 1B, Appendix A). The cluster most strikingly different from the others was “TP2”, which was composed of 47 geo-VRVs, each of which contained the D614 VRV at a maximum relative abundance of 100%, showing the early dominance of the D614 VRV in these states/territories. Clusters TP3 and TP5 included geo-VRVs of potential concern, as they included VRVs that appear to have emerged within the last few months in their specific states/territories. In contrast, most VRVs in the remaining clusters tended to expand and contract within relatively short times in a given state/territory, making such VRVs likely less important from a public health perspective. We termed these 35 VRVs that were uniquely identified in Clusters TP2, TP3, and TP5 as “pressing VRVs”.

### 2.2. Comparison with AA Positions Where Substitutions Have Been Identified within US-Circulating VOIs and VOCs

We next compared the 90 VRVs and the 35 pressing VRVs with the 12 and 13 AA positions that have been shown to harbor substitutions (AA-subs) within US-circulating VOIs and VOCs, respectively [2]. The 90 VRVs included nine and eight AA-subs in VOIs and VOCs, respectively; the 35 pressing VRVs included four and eight AA-subs in VOIs and VOCs, respectively (Figure 1C), even though all VOI/VOC sequences were excluded from the current analysis. Notably, 25 of the VRVs that have not been previously identified as an AA-sub in a VOI or VOC appear to have emerging trajectories, demonstrating the potential of the SLS method to identify novel Spike AA positions that may warrant further investigation/observation.

Five VOI/VOC AA-subs (Y144, F888, V1176, H69, K417) were not identified as a VRV. Appendix A shows the state/territory-specific relative abundances over time for states/territories where substitutions were identified at these five positions (albeit without meeting the statistical significance criteria for identification as a VRV). Our data suggest that, individually, these AA positions may be of less interest in the US.

### 2.3. Timely Detection of Emerging VRVs

Timely detection of potentially fast-emerging VRVs, and conversely, identification of VRVs likely not of concern, are both important for informing public health guidelines and for influencing research priorities. Given the importance of timely detection, we use the first time when the maximum proportion (Pmax) of a VRV exceeded 10% as the first reportable time (see the formula for calculating Pmax in Section 3.3.1 of Materials and Methods). For each out of the set of AA-subs within VOIs/VOCs that were also identified as VRVs, Table 1 compares within each state/territory the time of detecting an emerging VRV as calculated by the SLS method vs. the first appearance of the AA-sub in the scientific literature. The SLS method identified emerging VRVs in an average of 207 days, while the average of reported values in the literature was 299 days. E484, an AA-sub in the B.1.1.7, P.1, and B.1.351 variants, was an exception as it was not detectable in the US until day 370 when it was first detected as a VRV in Rhode Island.

### 2.4. VRV-Haplotypes

SARS-CoV-2 is a single-stranded (“haploid”) RNA virus. The presence of multiple VRVs found in a patient form a VRV-haplotype. The accumulation of multiple VRVs on a single RNA strand could affect protein function more than a single VRV. To identify VRV-haplotypes, we performed unsupervised learning of selected VRVs and cases through a two-way hierarchical cluster analysis state/territory-by-state/territory. As shown in Appendix A, some VRV-haplotypes are shared across states/territories, but most are not. Figure 2, for example, shows the results of the unsupervised case and VRV clustering for Washington state. The heatmap shows that multiple VRVs tend to aggregate among subsets of cases, the inspection of which can reveal VRV-haplotypes. For example, the case cluster “PG7”, which included 208 cases, had VRVs from the “RG4” and “RG3” clusters, which included the VRVs (G142-S155-F157-E180-K444-L452-T478-D614-Q677-P681-T732-T859-S940-D950) (see Appendix A).

Through algorithmic comparison and merging of different case clusters, six VRV-haplotypes (W1 through W6) were identified in Washington, while the “W6” cluster (7130 cases) carried only a single VRV, D614 (Table 2). Comparison across VRV-haplotypes suggested that W6 evolved to W3, W4, and W5 via the acquisition of an additional mutation at T732, Q677, and D178, respectively. Similarly, both W3 and W4 could have evolved to W2 via the acquisition of an additional mutation at Q677 or T732, respectively.

VRV-haplotype blocks are identified from unsupervised learning. Within each block, there can be multiple VRV-haplotypes that consist of polymorphic residues; individual VRVs may take either the reference residue or a substitution. For example, VRV-haplotype W1 had 10 haplotypes (Table 2), where the number after the hyphen indicates the number of substitutions. For example, the haplotype “ICRVNGA” had four substitutions, and was observed twenty times in Washington.

Table 2 also displays the VRV-haplotypes observed in New York (N1 through N7). The most frequent block, N2, had seven VRVs and 16 unique haplotypes. Block N1 only differed from Block N2 via the acquisition of the P681 VRV, and thus the two blocks are closely connected. Similarly, Block N4, which probably gave rise to Block N3, had 14 unique haplotypes, including “GSRGNH” (six substitutions), which was observed 455 times. Lastly, N5 probably arose from N6 via N7, and had the “PGHI” haplotype (observed 367 times).

### 2.5. Naming VRV-Haplotypes via PANGO Lineages

As all sequences corresponding to VOI/VOC were excluded, the strains with detected VRVs are not currently undergoing special monitoring or characterization. We were thus interested in naming the identified VRV-haplotypes and the PANGO lineages assigned by GISAID. To this end, we selected VRV-haplotype blocks including four or more pressing VRV mutations, resulting in eight VRV-haplotype blocks. Table 3 cross-tabulates these VRV-haplotypes by their assigned lineages. Of particular interest, viruses with the haplotype “KGHA” of T478-D614-P681-T732 were observed 2132 times, and 2029 of them were assigned to the strain B.1.1.222. It is natural to name the haplotype T478K-D614G-P681H-T732A as a B.1.1.222. Another noteworthy strain is B.1.234, which corresponds to “SVGHF” and “SVGHS” of G142-E180-D614-Q677-S940 with exceptionally high frequencies (353 and 262). The remaining VRV-haplotypes mostly correspond to B.1. Fourteen other strains were found in more than 10 occurrences and may also be of potential interest.

### 2.6. Impact of VRV Haplotypes on Viral Structure

The SLS method includes homology modeling of Spike mutations, to predict possible consequences on Spike structure/function and to guide laboratory research. Inspection of the temporal dynamics of the VRV-haplotypes may be useful for identifying VRVs of interest. We performed homology modeling on two potentially interesting VRV-haplotypes, one of which we refer to as “UK-VRV” (N501-A570-D614-P681-T716-S982-D1118, from the UK variant cluster B.1.1.7) and another VRV-haplotype, S13-W152-L452-D614 (from the US variant cluster (B.1.94; B.1.427; B.1.429)).

The D614G mutation observed in the UK-VRV haplotype has been associated with increased infectivity/transmissibility [22,23,24]. Cryo-electron microscopy structures have recently been reported [25,26] that reveal the structural consequences of this mutation and provide a plausible mechanistic explanation for the increased infectivity of D614G-carrying variants. The D614 VRV has predominated in all US cases for which sequence information is available in the TP2 cluster (Figure 1B, Appendix A). The N501Y mutation (present in the B.1.1.7 variant) is located in the receptor-binding domain (RBD) and has been reported to enhance binding affinity to the angiotensin-converting enzyme-2 [8,27]. N501Y has also been shown to reduce susceptibility to some neutralizing antibodies (nAbs), although the B.1.1.7 variant appears to remain susceptible to some extent to natural infection-acquired and vaccine-induced nAbs [8].

Of the five remaining VRVs in the UK-VRV haplotype, A570, T716, and S982 seem relatively benign in that mutations at these positions are already decreasing in certain states/territories (this trend is also true to some extent for N501Y). While this observation may simply reflect inadequate sequencing efforts in recent months, it may also indicate that mutations at these positions do not confer any fitness advantage to the virus.

The two remaining VRVs in the UK-VRV haplotype, P681 and D1118, are more intriguing. Mutations at these two sites, particularly at P681, appear to persist in multiple states/territories. The P681H mutation occurs in the S1/S2 cleavage segment of the Spike protein, which is typically not resolved in cryo-electron microscopy or X-ray diffraction experiments. Thus, we cannot speculate on potential structural consequences of this mutation. However, the continued presence of this mutation in many states and its location in the Spike protein S1/S2 cleavage segment is most interesting. It is plausible that the P681H mutation increases the flexibility of the S1/S2 cleavage segment, which might lead to enhanced cleavage and infectivity. However, there are no experimental studies to date that demonstrate a significant increase in S1/S2 cleavage for this mutant relative to other variants. Likewise, we are not aware of any reports that D1118H impacts transmissibility or morbidity, but the location of this mutation in the Spike protein trimer assembly (Figure 3A,B) suggests it could impact trimer assembly structure/stability/dynamics.

The US variants also carry the D614G mutation. The VRV-haplotype S13I-W152C-L452R (ICR-3) appeared in Fall 2020 and is rapidly becoming dominant in states on the West Coast, as well as appearing in selected southwestern and southeastern states (Figure 3C–F). The S13I and W152C mutations, which are situated in the N-terminal domain (NTD) of the Spike protein, have been implicated in escape from NTD-targeting monoclonal antibodies [28]. The L452R mutation is situated in the RBD; homology modelling of the RBD–ACE2 complex shows that while R452 does not directly contact ACE2, the guanidinium side chain of R452 is surface-exposed and thus could potentially impact nAb binding (Figure 3G). The L452R mutation was recently shown to reduce binding affinity to some RBD-targeting monoclonal antibodies, as well as to reduce susceptibility to nAbs [28]. Thus, structural modeling of mutations in the S13-W152-L452 VRV-haplotype yields results that are consistent with the temporal dynamics of this VRV-haplotype.

## 3. Materials and Methods

### 3.1. Spike AA Sequences

Spike AA sequences (genome position: 21563–25384) from 189,727 COVID-19 cases in the US and selected US territories, along with their associated metadata, were retrieved from GISAID [29] (https://www.gisaid.org/ accessed on 23 March 2021). Geographic origin (one of the 50 US states, Washington DC, Puerto Rico, or the Virgin Islands) was available for 189,284 of the sequences. For 443 of the cases, no US state/territory origin information was available. To ensure adequate sample size, Spike sequences from North Dakota, South Dakota, and the Virgin Islands were combined with these 443 sequences, forming an “Other States” category (728 sequences). Among them, 21,391 sequences were classified as a VOI or VOC (Appendix A). These sequences were excluded, leaving 167,893 sequences for analysis (see Appendix A for monthly case numbers by state/territory).

### 3.2. Sequence Alignment and Transformation to VRV Indicators

Spike protein sequences were aligned to the Wuhan reference sequence [30] using MAFFT [31], yielding a complete “rectangular residue sequence matrix”. Sequences with at least one AA mutation (compared to the reference) were identified, enabling transformation of the residue sequence matrix to a matrix of binary VRV (mutant) indicators. Monomorphic residues led to columns of zeros and were eliminated from further analysis. We use VRV in this work to refer to a single AA position that harbors a substitution. We reserve the term “variant” in this work for identified VOIs and VOCs.

### 3.3. Statistical Learning Strategy (SLS)

#### 3.3.1. Modeling VRV Temporal Dynamics

To model non-linear temporal dynamics, a generalized additive model (GAM) was used to regress the VRV indicator over sample collection time via the following probability model for the jth binary VRV indicator,
(1)$$\mathrm{Pr}\left({\mathrm{VRV}}_{j}=1|t\right)=\frac{1}{1+\exp\left[-\alpha -s_{j}\left(t\right)\right]}$$
where α is a constant coefficient and *s_j_*(*t*) is a non-linear function of time *t*, and both are estimated by the restricted maximum likelihood method [32]. Upon completing the estimation, the above function was used with the estimated coefficient and non-linear function to compute locally averaged proportions of each jth VRV, yielding a *p*-value that measures if the function *s_j_*(*t*) deviates from zero. Also produced is the maximum proportion as $\mathrm{Pmax}=\max[\mathrm{Pr}\left({\mathrm{VRV}}_{j}=1|t\right)]$. To correct for multiple comparisons, VRVs *p*-values were converted to q-values (false discovery rates) [33]. The function “gam” was used to fit the GAM and the function “qvalue” was used to compute q-values (R packages MGCV [34] and qvalue [33], respectively). The smoothing parameter k = 7 was chosen, and VRVs present in fewer than 10 sequences (or >90% of all sequences, as for D614), or whose span between emergence and disappearance from circulation was less than seven days were assigned a *p*-value (and q-value) of one.

Upon fitting the GAM, the fitted values were used as locally averaged VRV proportions daily from the first to the last reporting day. Computed proportions over time describe VRV temporal dynamics. A VRV is identified by both statistical significance (q-value < 0.01) and being present in a substantial proportion (Pmax > 10%).

#### 3.3.2. Visual Representation of Temporal Dynamics

Within-state/territory: Temporal dynamics of <8 VRVs within a given state/territory were visualized with a line plot. For visualizing temporal dynamics of ≥8 VRVs within a given state/territory, unsupervised learning was applied, grouping VRVs with similar temporal patterns. Results were visualized with a heatmap.

Spatially integrated: To visualize spatiotemporal VRV dynamics, all state-specific temporal dynamics were integrated and unsupervised learning (one-way hierarchical clustering with the Euclidean distance with weights in favor of recent temporal trajectories and the “ward.D2” agglomeration method) [35] was applied.

#### 3.3.3. Missing Residue Imputation

Given the physical proximity and constrained functionalities, residues are in high disequilibrium, forming haplotypes on a single-stranded RNA sequence (“VRV-haplotypes”) [36]. Haplotype structures enable imputation similarly to SNP imputation. Ten residues were selected to form a “imputation set”. For each selected VRV, its missing values were imputed based on their empirical haplotype frequencies. Remaining missing residues (if any) are denoted by a lower case “x”. While technically, a missing residue sequence value cannot be distinguished from an AA deletion, we conjectured that AA deletions would be rare in Spike and treated all such cases as missing values.

#### 3.3.4. VRV-Haplotypes

A viral strain harboring multiple VRVs is referred to as a “VRV-haplotype”. To identify VRV-haplotypes, unsupervised learning was used to organize both cases and VRVs through a two-way hierarchical analysis [35]. 

The two-way hierarchical analysis used the parallel function “parDist” with the “tanimoto” distance measure and the function “hclust” with “ward.D2” algorithm in the R package. VRV-haplotypes were identified separately within each state/territory. 

Hierarchically organized COVID-19 cases and VRVs are visually represented by a heatmap for every state/territory, which can be divided in grids by R geographic case groups (case1, case2, …, caseR) and by C VRV groups (vrv1, vrv2, …, vrvC). 

As a matrix of VRV indicators was used to identify blocks of multiple VRVs among subsets of cases, these blocks are referred to as VRV-haplotype blocks, since individual VRVs within blocks may vary. Within each block, actual residues were used to construct VRV-haplotypes for estimating haplotype frequencies and linking haplotypes with viral strains. For evaluating the relationship of VRV-haplotypes with known variants, variant information for case sequences was retrieved from GISAID [29]. 

#### 3.3.5. Homology Modeling of Selected Haplotype Mutants

After identifying specific Spike protein mutants of interest from VRVs and related VRV-haplotypes, standard homology modeling methods were applied to generate 3D models. Since all mutants examined in this study occurred in haplotypes harboring the D614G substitution, reference structures harboring this mutation (PDB codes 6ZWV; 7KRR; 7KRS) [25,26] were used as templates. All model building and analysis was performed using the ChimeraX interactive molecular graphics package (Resource for Biocomputing, Visualization, and Informatics at UC San Francisco, CA, USA) [37]. Mutated side chain conformations were generated from the backbone-dependent rotamer library of Shapovalov and Dunbrack [38] and selected to minimize steric clash with neighboring residues. No additional structural refinement was performed. For homology modeling using the Spike receptor binding domain (RBD)-ACE2 complex, PDB code 6M0J [39] was used.

## 4. Conclusions

The continuous evolution of SARS-CoV-2 has already impacted public health guidelines and research priorities, with the potential of even more clinically consequential variants still to emerge. Here, we leveraged a public data resource and described a statistical learning strategy for analyzing large, complex SARS-CoV-2 sequence datasets while incorporating temporal and spatial information. We provided detailed information on the emergence and persistence (or disappearance) of specific mutations in US states/territories, helping identify mutations that may warrant further observation/investigation. Our approach can be applied to other pathogens for which sufficient genomic surveillance data are available, generating important, statistically rigorous, and visually interpretable information for the biomedical research community, clinicians and public health officials. Our approach can also provide insight on the evolution of mutants and linkages with known viral strains.

By applying the SLS method to 167,893 US sequences not classified as any VOI/VOC, we identified 77 novel individual VRVs, including 25 pressing VRVs that appear to have emerged in the US. Among these pressing VRVs, the haplotype (T478-D614-P681-T732) links with the strain B.1.1.222 and the haplotype (G142-E180-D614-Q677-S940) links with the strain B.1.234, both of which do not correspond to any current VOI/VOC. Additionally of note, if the SLS method is applied to all US sequences, all circulating VOIs/VOCs are identified (results not shown). Indeed, timely discovery of novel VRVs is an important feature of the SLS method, which directly identifies emerging VRVs without relying on established classification of viral strains, as reliable classification of new strains by phylogenic analyses requires a sufficient number of accumulated viral sequence data. We should emphasize that our analyses were based on samples collected from January 2020 through March 2021, so it is not surprising that many of the VRVs we identified involve mutations associated most closely with the Alpha variant (B.1.1.7), which was predominant in the U.S. and world-wide during this time period. However, we note that our analyses also identified a haplotype N6 (Table 2) that exhibits the subsequently well-documented mutation P681R, characteristic of the Delta variant that has dominated infections world-wide during the latter half of 2021. During our analysis time period, this N6 haplotype was a minor, but statistically significant, fraction of the observed haplotypes in patient samples. Recent experimental studies [40,41] indicate that this mutation enhances S1/S2 segment cleavage and makes a major contribution to the enhanced infectivity observed for the Delta variant. The fact that our analysis protocol identified this specific mutation long before the Delta variant became clinically relevant in the U.S. appears to support the contention that our method may be useful for the early recognition of potentially important mutations.

As part of the assessment of immune correlates of protection, many randomized, placebo-controlled COVID-19 vaccine efficacy trials measure Spike protein sequences from symptomatic COVID-19 endpoint cases, and sometimes also from SARS-CoV-2 asymptomatic infections. Sieve analysis of these viral sequences can be conducted to assess whether and how vaccine efficacy depends on Spike protein sequence features, including differential vaccine efficacy across the levels of VRVs and of VRV-haplotypes [42]. The graphical tools proposed here for spatiotemporal tracking of VRVs and VRV-haplotypes can be useful for sieve analysis. Firstly, they help define and communicate the set of VRVs and VRV-haplotypes of study endpoint cases that have sufficient variability to be able to assess whether vaccine efficacy depends on the feature. For example, given that most vaccines use the Wuhan strain as the vaccine-insert, VRVs that meet our Pmax > 0.10 criterion would readily have the level of variability required for sieve analysis, whereas VRVs with Pmax < 0.02 would likely not. Secondly, including the assignment to vaccine or placebo as a factor in the unsupervised clustering graphics applied to the vaccine efficacy trial sequence data sets may help communicate results of sieve analysis. Thirdly, many of the vaccine efficacy trials have been offering the vaccine to placebo recipients, such that the placebo arm is lost and long term follow-up occurs only in individuals originally vaccinated or newly (deferred) vaccinated [43]. The graphical tools may be applied to track study participant vaccine breakthrough virus VRVs and VRV-haplotypes over time, and to similarly track VRVs and VRV-haplotypes in GISAID data bases of unvaccinated persons matched by geography and time. A comparison of these two tracking results may aid sieve analysis during the long-term follow-up period of the vaccine efficacy trials.

Evidence is mounting that neutralizing antibodies acquired by natural infection [44,45] or through vaccination [46,47,48,49] are a correlate of protection against COVID-19. Therefore, it will be critical to assess whether and how VRVs and/or VRV-haplotypes in the infecting strains impact neutralizing antibody titers attained by natural infection [50], as well as whether and how they impact neutralization sensitivity to vaccine-induced neutralizing antibodies [10] and/or monoclonal antibodies [51]. One possibility is that the graphical tools used here could annotate VRVs and VRV-haplotypes according to impact on neutralization. Moreover, a subset of sieve analyses is designed to restrict to VRVs and VRV-haplotypes that are known to impact neutralization response to the given vaccine under study, to improve power and to contribute to understanding neutralizing antibody-based correlates of protection. Once VRVs or VRV-haplotypes that impact vaccine efficacy are identified and their impact is quantified, this information can be applied to inform models for predicting vaccine efficacy against circulating virus populations and to aid optimization of vaccine strain selection.

A limitation of our approach is that it is constrained by intrinsic sampling limitations because all sequences were collected and contributed by laboratories (Appendix A) without consistent sampling protocols. Hence, despite the large size of our dataset, the analyzed sequences were not nationally representative. Further, it is important to interpret our results in terms of VRV proportions among reported sequence data, rather than incidences or prevalence of VRVs, in the absence of reliably estimated denominators. To overcome this limitation, public health agencies need to consider a uniformly developed surveillance protocol, to sequence COVID-19 cases from well-defined populations.

## Figures and Tables

**Figure 1 viruses-14-00009-f001:**
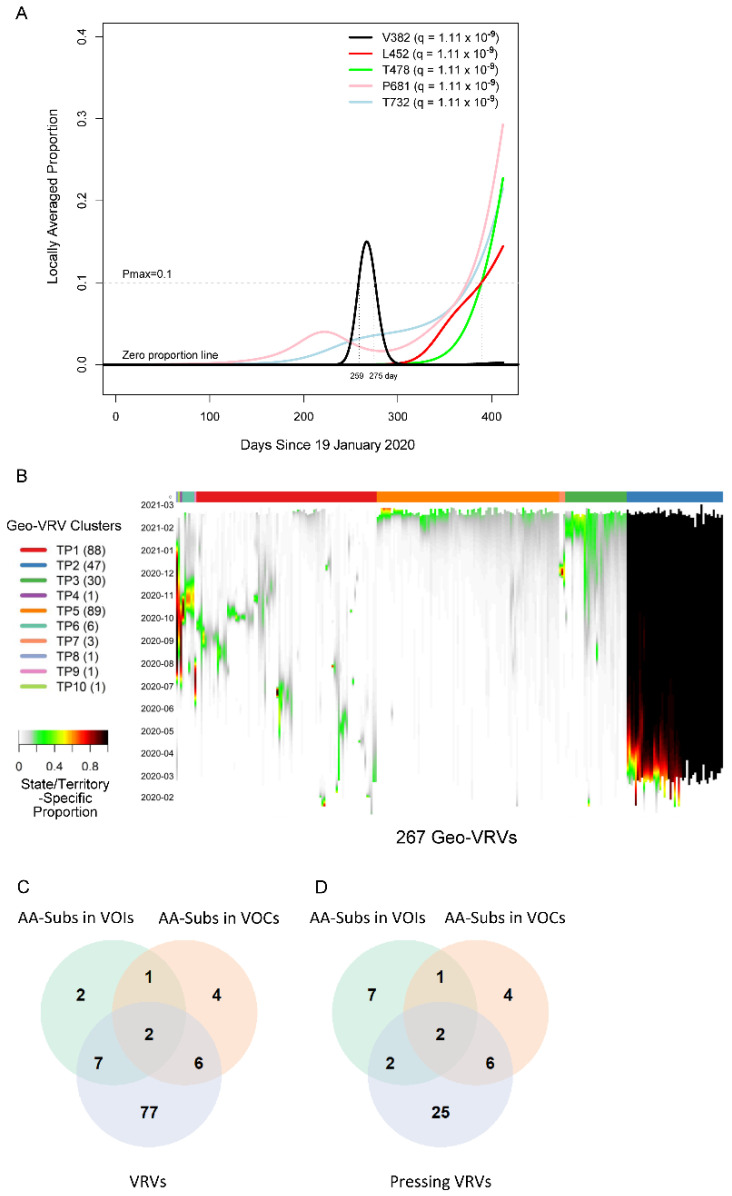
Viral residue variant (VRV) spatiotemporal patterns in the United States. (**A**) Locally averaged proportions over time of five VRVs (V382, L452, T478, P681 and T732), modeled using sequences from California. The horizontal gray dotted line denotes the maximum proportion (Pmax) cutoff of 10% (see the formula for calculating Pmax in Section 3.3.1 of Materials and Methods). V382 exceeded the Pmax cutoff of 10% on day 259 and dropped below the Pmax cutoff of 10% on day 275 (marked by the vertical gray lines). (**B**) Heatmap of the 267 identified geo-VRVs, with color designating the state/territory-specific VRV proportion at the sampling time as designated on the left-hand vertical axis. Geo-VRVs with similar temporal dynamics are grouped into 10 clusters (TP1 through TP10), as designated by the color bar at the top of the heatmap. (**C**,**D**) Venn diagrams showing the relationships between AA-subs in VOIs, AA-subs in VOCs, and (**C**) VRVs or (**D**) pressing VRVs. AA-subs, amino acid positions that have been shown to harbor substitutions within US-circulating variants; VOCs, variants of concern; VOIs, variants of interest.

**Figure 2 viruses-14-00009-f002:**
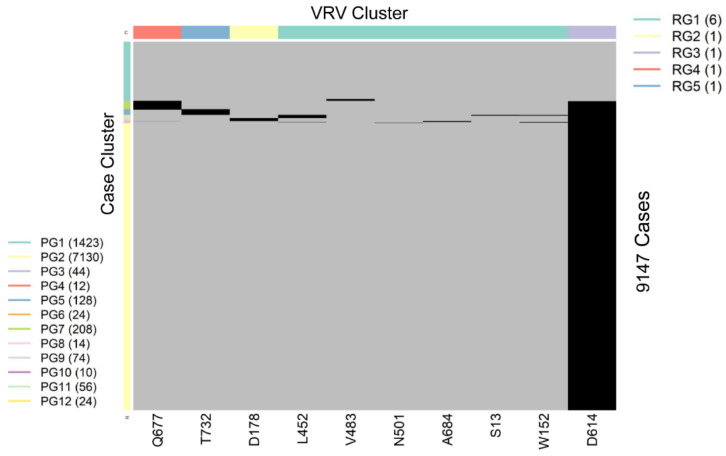
Heatmap showing the presence of 10 selected VRVs among 9147 cases in Washington state. Unsupervised learning was used to organize the 10 VRVs into 5 residue groups (RG1 through RG5) and to organize the 9877 cases into 12 patient groups (PG1 through PG12).

**Figure 3 viruses-14-00009-f003:**
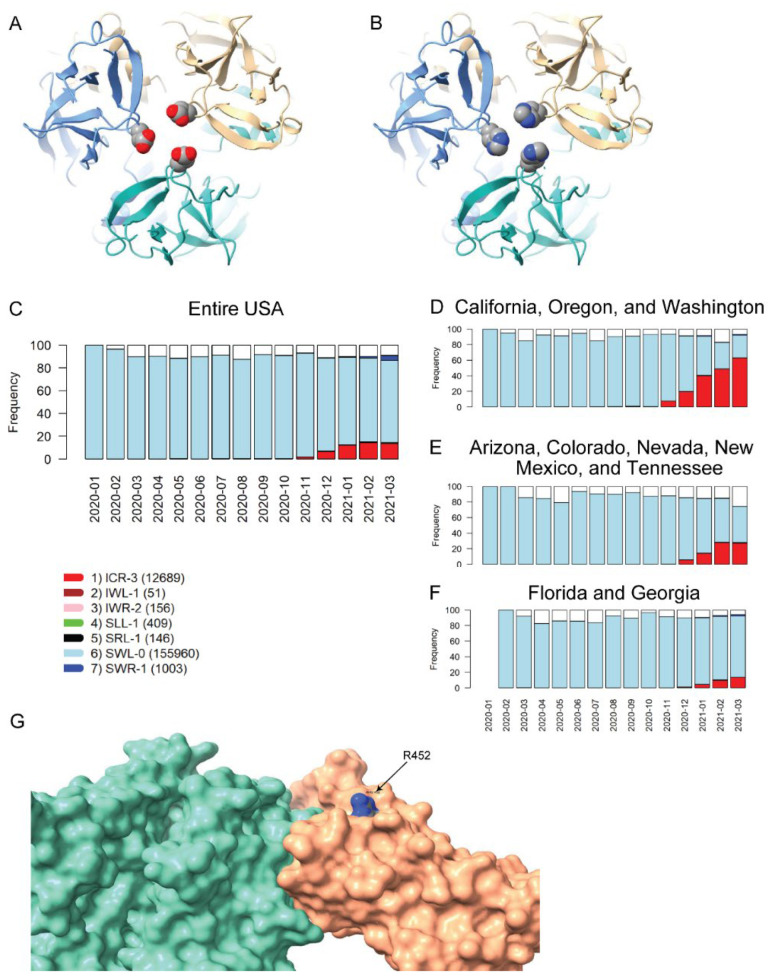
Homology modeling of Spike mutations and haplotypic polymorphisms over time of the S13-W152-L452 VRV-haplotype. (**A**,**B**) Modeled structure of the Spike protein trimer with (**A**) D1118 or (**B**) H1118 (homology-modelled using PDB entry 7KRS as the template structure). Spike protein monomers are displayed in blue, salmon, and aquamarine; aspartic acid and histidine residues are rendered as CPK images. (**C**–**F**) Frequencies over time for seven commonly observed haplotypic polymorphisms of the S13-W152-L452 VRV-haplotype, out of its polymorphisms in the US. Only haplotypic polymorphisms with at least 50 observations are included. Nomenclature is as follows: The first three letters designate the amino acids present at positions 13, 152, and 452, respectively; the number after the hyphen designates the number of amino acids at these three positions that do not match their reference strain equivalents. Numbers of sequences harboring each S13-W152-L452 haplotypic polymorphism (across the entire USA) are shown in parentheses. Frequencies of seven common S13-W152-L452 VRV-haplotypic polymorphisms (**C**) in the entire US; (**D**) in California, Oregon, and Washington combined; (**E**) in Arizona, Colorado, Nevada, New Mexico, and Tennessee combined; and (**F**) in Florida and Georgia combined. (**G**) Homology-modeled complex of the receptor-binding domain of the Spike protein (salmon), harboring the L452R mutation, bound to the angiotensin-converting enzyme 2 (ACE2) receptor (aquamarine). Within the R452 residue, nitrogen atoms are shown in blue and carbon atoms are shown in grey.

**Table 1 viruses-14-00009-t001:** Comparison by state/territory of the time to detect an emerging VRV by the SLS method vs. the first reported appearance of the AA-sub.

	L5	S13	V70	T95	W152	D253	L452	S477	E484	N501	A570	D614	Q677	P681	A701
Reporting Day *	301	301	301	301	301	87	301	331	87	362	362	362	362	362	362
Earliest SLS Detection Day Across All States	11	159	329	149	381	98	381	405	371	206	404	10	20	11	176
Alabama	63–186	-	-	-	-	-	-	-	404	-	-	63	253	-	-
Alaska	-	-	-	-	-	-	-	-	-	-	-	56	305–323	383	-
Arizona	-	-	-	-	-	-	-	-	-	-	-	26	357–400	353	-
Arkansas	-	-	-	-	-	-	-	-	-	-	-	56	329	-	-
California	-	398	-	-	402	-	390	-	-	-	-	45	-	374	-
Colorado	286	-	-	-	-	-	-	-	-	-	-	45	286	370	-
Connecticut	-	-	-	-	-	-	-	-	-	314	-	43	191	378	-
DC	-	-	-	-	-	-	-	-	391	-	-	47	-	344	-
Delaware	-	-	-	-	-	-	-	-	-	-	-	52	384	288	-
Florida	-	-	-	-	-	-	-	-	-	-	-	33	368	389	-
Georgia	-	-	-	-	-	-	-	-	-	-	-	41	345	407	-
Hawaii	175–190	-	-	-	-	-	-	-	-	-	-	46	374	174–376	-
Idaho	-	-	-	-	-	-	-	-	-	-	-	53	-	-	-
Illinois	-	-	-	-	-	-	-	-	-	-	-	24	366	380	-
Indiana	-	-	-	-	-	-	-	-	-	-	-	48	370	383	-
Iowa	-	-	-	-	-	-	-	-	-	-	-	48	273–388	-	-
Kansas	-	-	-	-	-	260–291	-	-	-	-	-	47	-	385	-
Kentucky	-	-	-	-	-	-	-	-	-	-	-	59	389–397	-	-
Louisiana	-	-	-	-	-	-	-	-	-	-	-	50	307	368	-
Maine	-	-	-	-	-	-	-	-	404	371	-	51	-	-	-
Maryland	-	-	-	-	407	-	394	-	390	-	-	45	-	230	-
Massachusetts	-	-	-	-	-	-	-	-	-	206	-	10	346	298	-
Michigan	-	-	-	149–177	-	-	-	-	-	264–273	-	50	361	-	-
Minnesota	186–294	-	-	-	-	-	-	-	-	-	-	46	297	387	-
Mississippi	144–215	-	-	-	-	-	-	-	-	-	-	42	353	392	-
Missouri	-	-	-	-	-	-	-	-	-	-	-	47	-	364–384	-
Montana	-	-	-	-	-	-	-	-	-	-	-	68	-	-	-
Nebraska	-	-	-	-	-	-	-	-	-	-	-	46	-	387	-
Nevada	-	392	-	-	396	-	393	-	-	-	-	37	391	394	-
New Hampshire	-	-	-	-	-	-	-	-	-	374	-	41	349–390	364	-
New Jersey	-	-	-	-	-	-	402	-	-	-	-	44	-	276	-
New Mexico	-	-	-	-	-	-	-	-	-	-	-	50	291	387	233–252
New York	11–13	-	-	-	-	-	386	-	414	-	-	11	-	11	-
North Carolina	-	-	-	-	-	-	-	-	-	-	-	44	-	382	-
North Dakota	-	-	-	-	-	-	-	-	-	-	-	57	328–363	-	-
Ohio	-	-	-	-	-	-	405	-	-	-	-	20	20	391	-
Oklahoma	-	-	-	-	-	-	-	-	-	-	-	54	306	-	-
Oregon	-	397	-	-	-	-	398	-	-	-	-	45	-	-	-
Pennsylvania	-	-	-	-	-	-	-	-	-	-	-	44	394	317	-
Puerto Rico	-	-	-	-	-	185–252	-	-	-	-	-	49	-	347	-
Rhode Island	-	-	-	-	-	-	-	405	371–384	356	-	40	358–398	379	-
South Carolina	-	-	-	-	-	-	-	-	-	-	-	46	405	368–389	-
Tennessee	50–141	-	-	-	-	-	-	-	-	-	-	50	318	-	-
Texas	-	-	-	-	-	-	-	-	-	-	-	23	360	378	-
Utah	-	159–173	-	-	-	98–190	-	-	-	-	-	44	-	358	-
Virginia	-	-	329–331	-	-	-	-	-	397	-	-	47	384	359	176–185
Washington	-	406	-	-	411	-	403	-	-	410	-	50	373	-	-
Wisconsin	-	-	-	-	-	-	409	-	-	265–271	-	12	299–407	411	-
Wyoming	-	381	-	-	381	-	381	-	-	-	-	51	-	392	-
Other States	-	-	-	-	-	-	-	-	393	404	404	34	-	368	375–381

For each of 15 amino acid positions shown to harbor a substitution in a VOI or VOC, the top two rows show the first reported date in the literature of a VOI or VOC harboring a substitution at the designated site vs. the date of VRV detection at the same amino acid position by the SLS method (across all states/territories. * “Reporting Day” was set to the 15th day in each month in which the relevant publication appeared. “SLS Detection Day” was set to the day at which the locally averaged proportion of the specific VRV exceeded 10% based on temporality models fitted in each states/territory. If the locally averaged proportion of the VRV later declined below 10%, the second day is shown after a dash. All numbers in the table express the number of days post-19 January 2021. A dash means that that VRV for that column was not detected in the state for that row. VOI, variant of interest; VOC, variant of concern.

**Table 2 viruses-14-00009-t002:** VRV-haplotypes identified in Washington and in New York: state-specific frequencies of cases, number of VRVs per VRV-haplotype, and haplotypic polymorphisms (state-specific frequencies). Unimputable residues are denoted with an “X”.

ID	VRV-Haplotype	Freq	L ^+^	Haplotypic Polymorphism-Number of Substitutions (Frequency)
Washington				
W1	S13-W152-L452-V483-N501-D614-A684	104	4	ICRVNGA-4(20)/IWRVNGA-3(4)/SCRVNGA-3(5)/SLLVNGA-2(5)/SRLVNGA-2(4)/SWLVTGA-2(4)/SWLVYDA-1(1)/SWLVYGA-2(5)/SWQVNGA-2(2)/SWRVNGA-2(54)
W2	D614-Q677-T732	12	3	GHS-3(11)/XXX-3(1)
W3	D614-T732	128	2	GA-2(126)/GI-2(2)
W4	D614-Q677	208	2	DH-1(9)/GH-2(110)/GP-2(89)
W5	D178-D614	74	2	GG-2(70)/NG-2(4)
W6	D614	7130	1	G-1(7125)/N-1(5)
New York				
N1	L5-L54-E132-Y453-T478-E484-D614-P681-T732	172	9	LLEYKEGHA-4(168)/LLEYKEGHT-3(4)
N2	L5-L54-E132-Y453-T478-E484-D614-T732	651	8	FLEYREGT-3(4)/FLEYTEDT-1(11)/FLEYTEGA-3(3)/FLEYTEGS-3(1)/FLEYTEGT-2(266)/FLEYTKGA-4(1)/FLEYTKGT-3(44)/LLEYKEGT-2(3)/ LLEYTAGT-2(1)/LLEYTEGA-2(51)/LLEYTEGI-2(2)/LLEYTEGS-2(24)/LLEYTKGS-3(2)/LLEYTKGT-2(171)/LLEYTQGT-2(8)/LLQYTEGT-2(59)
N3	D80-F157-L452-D614-P681-T859-D950	132	7	DFLGHID-3(108)/DFLGPID-2(18)/DFLGPNH-3(4)/DSLGPNH-4(2)
N4	D80-F157-L452-D614-T859-D950	637	6	DFQGND-3(4)/DFRGID-3(15)/DFRGNH-4(1)/DFRGTD-2(120)/DFRNTD-2(2)/DSRGNH-5(3)/DSRGTD-3(2)/GFRGND-4(1)/GFRGNH-5(1)/GSLGNH-5(9)/GSRGND-5(10)/GSRGNH-6(455)/GSRGNY-6(1)/GSRGTD-4(13)
N5	S494-D614-P681-T716	514	4	PGHI-4(367)/PGHT-3(55)/PGPT-2(52)/SGHI-3(19)/SGHT-2(8)/SGPI-2(13)
N6	D614-P681	1161	2	GH-2(1124)/GL-2(4)/GR-2(32)/GS-2(1)
N7	D614	10822	1	D-0(1)/G-1(10821)

^+^ Number of substitutions in VRV-haplotypes from the reference amino acids.

**Table 3 viruses-14-00009-t003:** VRV-haplotypes. Cross-tabulation of individual VRV-haplotypes with GISAID-assigned lineages in all 167,893 sequences, excluding lineages with fewer than 10 occurrences. “Freq”, corresponding haplotype frequencies; “Unknown”, sequences not assigned to any lineage.

Hap-Load	Freq	Unknown	A.2.4	B.1	B.1.1	B.1.1.1	B.1.1.171	B.1.1.222	B.1.1.29	B.1.1.304	B.1.1.317	B.1.152	B.1.165	B.1.166	B.1.2	B.1.215	B.1.234	B.1.256	B.1.324	B.1.350	B.1.354	B.1.360	B.1.399	B.1.94
(1) D80-F157-L452-D614-T859-D950																								
DSRGNH-5	63			58								5												
GSLGNH-5	9			9																				
GSRGND-5	21			19																			1	
GSRGNH-6	539			522					1			2		3									5	
(2) D80-S155-F157-L452-T859-D950																								
DRSRNH-5	39			39																				
GRSRND-5	3			3																				
GRSRNH-6	30			30																				
GSSRNH-5	509			492					1			2		3									5	
(3) G142-E180-D614-Q677-S940																								
SEGHF-4	3														1		1			1				
SVGHF-5	353																353							
SVGHS-4	273	2		1													262			8				
(4) S155-F157-L452-T859-D950																								
RSRND-4	3			3																				
RSRNH-5	69			69																				
SSRNH-4	533			511					1			7		3									5	
(5) S13-W152-L452-D614																								
ICLG-3	43	1		36											3									
ICRG-4	795	51		557									1	4	10			14			10	34	2	72
IWRG-3	120	1		77											7						2			28
SCRG-3	30	4		16											4									
(6) S494-D614-P681-T716																								
PGHI-4	521			467	1									1	1	20				3				
PGHT-3	194			100	8			3					31		2	3			29				1	
RGHI-4	3															3								
SGHI-3	38			19	3			1							4									
(7) T478-D614-P681-T732																								
KGHA-4	2132	11		17	2		14	2029	18	1	12				2									
KGHS-4	6																							
KGHT-3	159			4	57		3		67	8												1		
KGPA-3	5			1				3	1															
TGHA-3	85			13				63	2	1	2				2									
(8) F157-L452-D614-T859																								
FQGN-3	22					22																		
FRGI-3	15		14	1																				
FRGN-3	5			5																				
SLGN-3	11			10											1									
SRGN-4	625			601					1			7		3									6	
SRGT-3	37			33																				

Green shading: >100 occurrences. Light green shading: >10 occurrences.

## Data Availability

All sequence data analyzed here are publicly available at GSIAD (https://www.gisaid.org/). For this analysis, data were retrieved on 23 March 2021.

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
