# Peer review of "Tracking SARS-CoV-2 Spike Protein Mutations in the United States (January 2020—March 2021) Using a Statistical Learning Strategy"

_viruses, 2021, doi:10.3390/v14010009_

Round 1

Reviewer 1 Report

Many of the sites of interest noted in this paper are found in the "Delta variant" and sublineages subsequently derived from Delta (B.1.617).  It is important to know which mutations are carried by a lineage which is more infectious and thus rapidly increasing.  The "delta variant" and others are not mentioned in the paper, other than to note the B.1.617 lineages as variants of interest in the introduction.

This papers notes which sites in Spike (such as D614 or P681) are detected as changing, but does not note which amino acid they changed to (D614G, P681R).  It is important to note the new amino acid. 

Page 15 discusses the P681H mutation, which impact furin cleavage, although perhaps not as much as the P681R amino acid change.  P681H is noted in several major lineages of SARS-CoV-2.

Author Response

Many of the sites of interest noted in this paper are found in the "Delta variant" and sublineages subsequently derived from Delta (B.1.617).  It is important to know which mutations are carried by a lineage which is more infectious and thus rapidly increasing.  The "delta variant" and others are not mentioned in the paper, other than to note the B.1.617 lineages as variants of interest in the introduction.

Response: Thank you for the question. Indeed, genomic surveillance using phylogenic analysis – such as classification of new viral sequences to existing lineages – plays an important role in tracking the emergence and spread of new variants, as we discuss in our Introduction. In contrast, our statistical learning strategy is fundamentally different from such “lineage-based approaches” and provides an important complement, without relying on assignment of any lineage. For this reason, terminology such as “the delta variant” is not relevant to the bulk of our paper. However, it is interesting to note that many VRVs associated with Delta have been found, prior to formal designation of the Delta variant.  Again, the reviewer’s comment highlights the power of the statistical learning strategy.  This point is now indicated in Discussion; see lines 299-302 of the revision:

“Indeed, timely discovery of novel VRVs is an important feature of the SLS method, which directly identifies emerging VRVs without relying on established classification of viral strains, since reliable classification of new strains by phylogenic analyses require a sufficient number of accumulated viral sequence data.”

This papers notes which sites in Spike (such as D614 or P681) are detected as changing, but does not note which amino acid they changed to (D614G, P681R).  It is important to note the new amino acid. 

Response: We appreciate this point, and agree that indicating which amino acid the reference amino acid is mutated to can be helpful.  During our exploration of Spike protein data, however, we found that some reference amino acids exhibit point mutations to multiple different amino acids.  For example, P681 is commonly mutated to H, but is also mutated in some cases to L, R and S (see polymorphic amino acids in corresponding VRV-haplotypes in Table 2).  Therefore, conventional nomenclature is inadequate, especially, for describing VRV-haplotypes.  Fortunately, the reference amino acid with the specific position uniquely specifies the amino acid of interest, and is thus appropriate for reproducible analysis of protein sequence data.  Please see lines 91-93 of the revision:

“Note that we denote the VRV of interest using the reference amino acid position alone (e.g. P681), given that some reference amino acids show point mutations to more than one amino acid (see examples below).”

Page 15 discusses the P681H mutation, which impact furin cleavage, although perhaps not as much as the P681R amino acid change.  P681H is noted in several major lineages of SARS-CoV-2.

Response: This is an important point and a useful suggestion from the reviewer. The P681H mutation was discussed primarily because it was so prevalent in most of the VRV-haplotypes we identified, and also because it occurs in the important S1/S2 cleavage segment. As the reviewer notes, there is no experimental evidence to indicate that P681H has a major impact on furin cleavage, certainly not as dramatic as that observed for the P681R mutation associated with the Delta variant that has dominated in recent months. We did not discuss the P681R mutant explicitly in this manuscript because our analyses were based on sequence data collected from January, 2000, through March, 2021, and the Delta variant was not yet clinically relevant in the U.S. during that time period. We do note that our analyses did reveal that the P681R mutation was beginning to appear during this earlier time period, and we have added some discussion to clarify and emphasize this point per the reveiwer’s suggestion. Please see lines 302-313 of the revision:

“We should emphasize that our analyses were based on samples collected from January 2020 through March 2021, so it is not surprising that many of the VRVs we identified involve mutations associated most closely with the Alpha variant (B.1.1.7), which was predominant in the U.S. and world-wide during this time period. However, we note that our analyses also identified a haplotype N6 (Table 2) that exhibits the subsequently well-documented mutation P681R, characteristic of the Delta variant that has dominated infections world-wide during the latter half of 2021. During our analysis time period, this N6 haplotype was a minor, but statistically significant, fraction of the observed haplotypes in patient samples. Recent experimental studies (29) indicate that this mutation enhances S1/S2 segment cleavage and makes a major contribution to the enhanced infectivity observed for the Delta variant. The fact that our analysis protocol identified this specific mutation long before the Delta variant became clinically relevant in the U.S. appears to support the contention that our method may be useful for early recognition of potentially important mutations.”

Reviewer 2 Report

The authors propose a statistical learning strategy to identify and follow potentially relevant mutations of SARS-COV-2 spike protein. Analysis of US spike sequences deposited in GISAID data base revealed several new variants. Most known variants of interest and variants of concern were identified by the learning strategy described in the manuscript several days earlier than by the genomic surveillance system used in the US. Moreover VRV-haplotypes were described and the potential functional impact of the mutations was addressed by structural modeling.

  • One of the results is that SLS is much faster in the identification of new variants than standard methods used for viral genome surveillance. What do you think are the main reasons for this?
  • Reference list must be updated. Reference 30 („in press“), for example, is already published.
  • Table 1: Title should be more comprehensible
  • Table 2: Abbreviations should be explained, e.g.(L). What means L?
  • Figure S3 legend should describe what means TP
  • Figure S4 legend should explain also RG and PG
  • Figure 2: the case cluster PG8, which was given as an example is not or only hardly visible
  • Page 11 line 175: which case clusters do the authors mean?
  • Table 3: column headers in the top row are cut
  • The reviewer is confused by the nomenclature: Does W1 to W6 from table 2 resemble WA1 to WA6 in Table S6? And does W1 and W2 on page 14 is the same as W1 and W2 in table 2? Please clarify and make it clear in the manuscript!
  • The abbreviation nAbs should be introduced.

Author Response

  • One of the results is that SLS is much faster in the identification of new variants than standard methods used for viral genome surveillance. What do you think are the main reasons for this?

Response:  The primary reason is associated with the assignment of new strains.  For phylogenic analysis to identify new strains, it is necessary for such emerging strains to accumulate a sufficient number of mutations on a sufficiently large number of viruses.  In contrast, SLS bypasses the need for strain assignment, and directly assesses emergencies of one or multiple SNVs.  Hence, SLS could identify new SNVs earlier than phylogenic approaches.  This point is noted in the revision (see lines 299-302 as discussed in the response to Reviewer 1).

  • Reference list must be updated. Reference 30 („in press“), for example, is already published.

Response: We have updated all citations of preprints to their corresponding published articles when applicable. We have also filled in volume/page information for articles that were lacking this information when originally cited in an early-online format. All references are now current.

  • Table 1: Title should be more comprehensible

Response: We have changed to “Comparison by state/territory of the time to detect an emerging VRV by the SLS method vs. the first reported appearance of the AA-sub.”

  • Table 2: Abbreviations should be explained, e.g.(L). What means L?

Response: We have added an explanation for this abbreviation L as “Number of substitutions in VRV-haplotypes from the reference amino acids” to the table.

  • Figure S3 legend should describe what means TP

Response: TP stands for temporal pattern.  Following the clustering analysis, we identified ten different temporal patterns, labeled “TP1” through “TP10”. These labels, along with the vertical color bar on the left side of the plot, refer to the 10 individual State-VRV clusters identified in the plot. We have edited the Fig. S3 legend accordingly.

  • Figure S4 legend should explain also RG and PG

Response: We have added “The “PG” (“patient group”) and “RG” (“residue group”) labels in the lower left corner of each plot correspond to the row (case) and column (VRV) clusters identified by two-way hierarchical cluster analysis, also identified by the vertical and horizontal color bars at the left and top, respectively, of each plot.”

  • Figure 2: the case cluster PG8, which was given as an example is not or only hardly visible

Response: This is a great point. We have chosen a different case cluster to highlight in the text, which has more cases and is thus more visible. The revised text reads as follows: “For example, the case cluster “PG7”, which includes 208 cases, has VRVs from the “RG4” and “RG3” clusters, which include the VRVs (G142-S155-F157-E180-K444-L452-T478-D614-Q677-P681-T732-T859-S940-D950) (See Table S5).” 

  • Page 11 line 175: which case clusters do the authors mean?

Response: In the SLS strategy, we apply a numerical algorithm to compare case clusters with respect to their number of different substitutions, and merge clusters by pre-established criteria.

We realize that the sentence referred to by the reviewer could be edited to improve clarity. We have deleted “Collectively, these four case clusters…from the “RG3, “RG4”, and “RG5” clusters” and instead added a more general description, as follows:

“Through algorithmic comparison and merging of different case clusters…” (line 176)

  • Table 3: column headers in the top row are cut

Response: Corrected.

  • The reviewer is confused by the nomenclature: Does W1 to W6 from table 2 resemble WA1 to WA6 in Table S6? And does W1 and W2 on page 14 is the same as W1 and W2 in table 2? Please clarify and make it clear in the manuscript!

Response: Thank you for pointing this out. For the first question, Table 2 shows VRV-haplotypes identified in Washington and in New York, whereas Table S6 shows results from a subset of VRVs (pressing VRV-haplotypes).  So for clarity, we have removed Table S6 and the related text from the revision.

For the second question, no, W1 and W2 on page 14 are not the same as W1 and W2 in Table 2. We see how this could be confusing. We have changed W1 to “UK-VRV” and omitted the “W2” designation, since the W2 VRV-haplotype is not discussed further in the text. See lines 216-219 in the revision:

“We performed homology modeling on two potentially interesting VRV-haplotypes, one of which we refer to “UK-VRV” (N501-A570-D614-P681-T716-S982-D1118, from the UK variant cluster B.1.1.7) and another VRV-haplotype, S13-W152-L452-D614 [from the US variant cluster (B.1.94; B.1.427; B.1.429)].”

  • The abbreviation nAbs should be introduced.

Response: Done.

Reviewer 3 Report

Reviewer #1:

Coronavirus disease 2019 (COVID-19) has emerged as a new world pandemic, infecting millions of people with a substantial mortality. There is significant interest study to analyzed the tracking SARS-CoV-2 Spike Protein Mutations in the United States.

Recently publications show several data with several SARS-CoV-2 viral variants and how these affect the vaccine efficacy in this review could be help in understand this pandemic in the context S protein mutations.

In this manuscript, by Lue Ping et al titled " Tracking SARS-CoV-2 Spike Protein Mutations in the 2 United States (2020/01 – 2021/03) Using a Statistical Learning 3 Strategy".

The authors performed an analysis identified several VRV-haplotypes that have not observed in a VOI/VOC. Also, structural modeling showed the potential functional impact of D1118H and L452R mutations.

There are minimal concerns that to be addressed.

This manuscript is well written and sites key findings in the field, therefore it will be helpful for epidemiological investigators in evolution (S mutations) research. The study would benefit the section on next vaccine generation.

Comments to improve the clarity of the manuscript are provided below.

Comments for the authors' consideration:

  1. Please add a section with these VRV cluster in other continents or other US states.

Author Response

  1. Please add a section with these VRV cluster in other continents or other US states.

Response: We agree that it is important to evaluate VRV clusters in other countries.  In fact, we have done some preliminary analyses on this topic, yielding many interesting distinct patterns.  However, we feel it would be difficult to constrain the reporting of such results to a small section, and conversely including a larger section on these results would somewhat sacrifice the main messages of the current work.  Thus, we believe that such analyses are best suited to a follow-up paper, and are out of scope of the present work.

Round 2

Reviewer 1 Report

The P681H mutation should be noted for as a charge change in the furin cleavage region somewhat similar to P681R.  Although as far as I know it is still not proven to increase cleavage, it seems likely based on charge more than on flexibility.  Both H and R at 681 seem to be positively selected relative to P, in several lineages.

Recombination between lineages, or between virions of the same lineage with only minor changes, is not mentioned in this paper.  However it seems to be increasingly evident that recombination is playing a significant role in SARS-CoV-2 evolution.  The authors should mention that recombination can invalidate or confound phylogenetic analyses, and this unsupervised learning method is a useful alternative.  I don't think it is necesssary to change the current text of the paper for this, if that will delay publication.  But the authors should consider the aspect of their work.